# Homeobox Gene Expression Dysregulation as Potential Diagnostic and Prognostic Biomarkers in Bladder Cancer

**DOI:** 10.3390/diagnostics13162641

**Published:** 2023-08-10

**Authors:** Fee-Wai Chin, Soon-Choy Chan, Abhi Veerakumarasivam

**Affiliations:** 1Department of Biomedical Sciences, Faculty of Medicine and Health Sciences, Universiti Putra Malaysia (UPM), Serdang 43400, Selangor, Malaysia; feewaic@gmail.com; 2School of Liberal Arts, Science and Technology, Perdana University, Kuala Lumpur 50490, Malaysia; 3School of Medical and Life Sciences, Sunway University, Bandar Sunway 47500, Selangor, Malaysia

**Keywords:** biomarker, bladder cancer, diagnosis, homeobox genes, prognosis, tumorigenesis

## Abstract

Homeobox genes serve as master regulatory transcription factors that regulate gene expression during embryogenesis. A homeobox gene may have either tumor-promoting or tumor-suppressive properties depending on the specific organ or cell lineage where it is expressed. The dysregulation of homeobox genes has been reported in various human cancers, including bladder cancer. The dysregulated expression of homeobox genes has been associated with bladder cancer clinical outcomes. Although bladder cancer has high risk of tumor recurrence and progression, it is highly challenging for clinicians to accurately predict the risk of tumor recurrence and progression at the initial point of diagnosis. Cystoscopy is the routine surveillance method used to detect tumor recurrence. However, the procedure causes significant discomfort and pain that results in poor surveillance follow-up amongst patients. Therefore, the development of reliable non-invasive biomarkers for the early detection and monitoring of bladder cancer is crucial. This review provides a comprehensive overview of the diagnostic and prognostic potential of homeobox gene expression dysregulation in bladder cancer.

## 1. Introduction

Homeobox genes are a superfamily of regulatory genes that encode transcription factors that regulate cell differentiation and morphogenesis during embryogenesis [1,2]. A homeobox gene contains a homeobox sequence that encodes a homeoprotein that contains a highly conserved homeodomain. The homeodomain acts as a transcriptional regulator to activate and repress downstream target genes [3,4]. Homeobox genes can be categorized into clustered (HOX) and non-clustered (non-HOX) homeobox genes. HOX homeobox genes are further classified into four gene clusters (HOX A, HOX B, HOX C, and HOX D) that are located on different chromosomes. Non-HOX homeobox genes are dispersed throughout the genome and are located outside the HOX clusters [5].

The dysregulation of homeobox genes has been reported in various human cancers such as bladder, breast, colorectal, gastric, kidney, liver, lung, ovarian, and prostate cancers [6,7,8,9,10,11,12,13,14]. The dysregulated expression of homeobox genes in cancers has been identified in various stages of tumorigenesis that include epithelial–mesenchymal transition (EMT), cell proliferation, invasion, and metastasis [15,16,17,18]. The dysregulation of homeobox genes is driven by several mechanisms, such as DNA hypermethylation, loss of heterozygosity, gene amplification, and histone deacetylation [19]. Interestingly, a homeobox gene may have tumor-promoting or tumor-suppressing properties depending on the cell lineage or specific organ in which it is expressed [16,17,18].

Bladder cancer is the most common urinary tract cancer, affecting men more frequently than women [20,21]. Urothelial cell carcinoma (UCC) is the most common histological type; accounting for about 90% of all bladder cancers [22]. Approximately 70% of UCCs are non-muscle invasive bladder cancers (NMIBCs; carcinoma in situ, stages Ta and T1), while the remaining 30% are muscle invasive bladder cancers (MIBCs; stages T2-T4) [23]. Up to 50–70% of patients with NMIBC develop tumor recurrence after transurethral resection of the bladder tumor (TURBT). Of these, 10–20% of the patients will progress to MIBC [24,25]. Unfortunately, patients with MIBC frequently progress to become metastatic cancer and when this happens, patients have the poorest prognosis with a 5-year survival rate of less than 5% [26].

The high recurrence and progression rate of bladder cancer necessitate frequent and long-term surveillance [27]. However, the use of cystoscopy for the diagnosis and surveillance of bladder cancer is invasive. The procedure often causes pain and bleeding in patients, resulting in moderate to low compliance towards surveillance follow-up [28]. Thus, the development of reliable and effective biomarkers for bladder cancer is crucial. Therefore, this review aims to provide a comprehensive review of the existing literature related to the dysregulated expression of homeobox genes in bladder cancer and assessing the diagnosis and prognosis biomarker potential in bladder cancer.

## 2. Method of Search

A comprehensive literature search was conducted in PubMed search engine for relevant articles published as of 24 March 2022. Search was performed using the “All Fields” search function based on the search terms of “homeobox gene symbol” and “bladder cancer” used in combination with the Boolean operator “AND”. Studies regarding the following topics were selected: homeobox genes in bladder cancer related to (1) gene expression analysis, (2) transcriptomic analysis, (3) phenotypic assay, (4) microRNAs (miRNAs), (5) long non-coding RNA (lncRNA), and (6) DNA methylation. Studies regarding the following topics were excluded: (1) single nucleotide polymorphisms (SNPs), (2) mutations, (3) gene fusion, (4) alternative splicing, and (5) other cancers (not bladder cancer). In addition, journal articles written in (1) non-English, (2) case reports, and (3) review articles were also excluded.

## 3. Homeobox Genes in Bladder Cancer

A comprehensive summary of the literature on the dysregulation of homeobox genes expression in bladder cancer is presented in this review. A schematic representation of the dysregulated homeobox genes in bladder cancer is shown in Figure 1. It was reported that some of these homeobox genes were associated with bladder tumorigenesis (Figure 2) and clinical outcomes of bladder cancer (Figure 3).

### 3.1. The ANTP Homeobox Class

#### 3.1.1. *CDX2*

The *CDX2* encodes a transcription factor that is involved in the differentiation and proliferation of intestinal epithelial cells [29,30,31]. Differential CDX2 expression has been reported in different subtypes of bladder cancer. Positive CDX2 expression was detected in 0–22% of urothelial carcinoma of the bladder (UCB) [32,33], in 50% of UCBs following augmentation cystoplasty [34], in 85% of urachal adenocarcinomas, and in 100% of urachal carcinomas [33]. The *CDX2* could be a useful biomarker to distinguish some subtypes of bladder cancers that have difficulty in differential diagnosis due to their shared morphologic resemblance. CDX2 expression was detected in 47% of primary bladder adenocarcinomas but was not detected in any primary or secondary colorectal adenocarcinomas. These findings suggest that *CDX2* is a promising biomarker to distinguish primary bladder adenocarcinomas from the morphologically similar colorectal adenocarcinoma [35].

Intestinal metaplasia of the bladder exhibited a different and contrasting CDX2 immunoprofile as compared to that of its morphologically similar cystitis glandularis of the bladder. Nuclear CDX2 staining was observed in 83% of the intestinal metaplasia, whereas it was not observable in the cystitis glandularis [36]. There is a difficulty in differential diagnosis between urothelial carcinoma in situ with glandular differentiation (CISg), and concurrent conventional urothelial carcinoma in situ (cCIS). However, CDX2 expression was detected in 76% of the CISg but was not detected in the cCIS. These findings suggest that CISg can be distinguished from cCIS by immunohistochemistry (IHC) using CDX2 as the diagnostic biomarker [37].

#### 3.1.2. *EN1* and *HOXB9*

The Database for Annotation, Visualization, and Integrated Discovery (DAVID) analysis for differentially expressed genes (DEGs) in bladder cancer tissues as compared to normal bladder tissues identified *EN1* and *HOXB9* as key genes that are involved in the development and progression of bladder cancer [38].

#### 3.1.3. *EN2*

Majority of bladder cancer cell lines expressed higher *EN2* mRNA levels compared with the normal urothelium cell line (NHU-BTERT). These bladder cancer cell lines (EJ, RT4, RT112, and UMUC3) had moderate to high *EN2* expression, whereas the normal urothelium cell line had low *EN2* expression. EN2 protein was expressed in UCC, squamous cell carcinoma (SCC), and adenocarcinoma. In contrast, EN2 protein was not expressed in normal bladder tissue [39].

*EN2* mRNA expression was significantly higher in bladder cancer cells than in the human uroepithelial cells (SV-HUC-1). The overexpression of *EN2* promoted cell proliferation and invasion in vitro but inhibited apoptosis. Meanwhile, the knockdown of *EN2* significantly promoted cell cycle arrest and apoptosis but inhibited cell proliferation and invasion in vitro as well as decreased tumor growth in vivo. Moreover, *EN2* knockdown significantly decreased PI3K, pAkt-308, and pAkt-473 expression but increased *PTEN* expression. These findings suggest that *EN2* induced the activation of the PI3K pathway by inhibiting *PTEN*. Taken together, *EN2* may be a candidate oncogene that plays a crucial role in the development and progression of bladder cancer. In addition, *EN2* may be a potential therapeutic target for bladder cancer [40]. The upregulation of *EN2* and the downregulation of miR-27b were found in bladder cancer tissues and cell lines. *EN2* is a direct target of miR-27b in bladder cancer, wherein *EN2* is negatively regulated by miR-27b. The miR-27b significantly inhibited cell proliferation and invasion. It also enhanced the apoptosis of bladder cancer cells. The overexpression of *EN2* reversed the biological effects of miR-27b on bladder cancer cells. These findings suggest that *EN2* is involved in the development of bladder cancer through the negative modulation of miR-27b [41].

In urine samples, patients with stage ≥T2 tumors had significantly higher urinary EN2 protein concentrations as compared to those with stage Ta and T1 tumors. In addition, higher grade (grades 2 and 3) tumors were associated with higher mean urinary EN2 protein concentrations as compared to low grade (grade 1) tumors. These findings suggest that *EN2* plays a role in the pathogenesis of bladder cancer. Urinary EN2 was detected in the majority of NMIBCs, with a sensitivity of 82% and a specificity 75%. Therefore, urinary *EN2* protein could be used as a sensitive and specific biomarker for NMIBC to reduce the use of cystoscopy for bladder cancer surveillance [39]. A rapid test prototype that utilized lateral flow immunoassay technology was successfully developed to screen urine samples for the detection of EN2 protein. This non-invasive in vitro diagnostic test had an 85.5% sensitivity and a 71.7% specificity and was deemed to surpass the sensitivity and specificity of many commercially available bladder cancer markers and tests [42]. This further supports the diagnostic role of *EN2* in bladder cancer. In addition, two multi biomarker panels that include *EN2* were identified as potential non-invasive diagnostic tools for the diagnosis of recurrent NMIBC [43,44]. Taken together, the use of *EN2* as a diagnostic biomarker is promising.

#### 3.1.4. *HOXA1*

The mRNA expression levels of *HOXA1* and *KDM3A* were significantly higher in bladder cancer tissues as compared to normal bladder tissues. *KDM3A* was found to regulate the growth of bladder cancer cells and *HOXA1* expression through the demethylation of H3K9me2. The knockdown of *KDM3A* downregulated *CCND1* (cell cycle progression factor) expression and arrested G1/S transition of cell cycle. The upregulation of *HOXA1* activated *CCND1* transcription and promoted the G1/S transition. Meanwhile, the knockdown of *HOXA1* suppressed the growth of bladder cancer cells. These findings indicate that elevated *KDM3A* expression in bladder cancer cells transactivates *HOXA1* expression through the demethylation of H3K9me2 [45].

#### 3.1.5. *HOXB2*

The methylation of *HOXB2*, *FRZB*, and *KRT13* was greater in invasive bladder cancers as compared to non-invasive bladder cancers. An increased *HOXB2* methylation was significantly associated with invasive bladder cancer, with an 8.6-fold increased risk of developing invasive bladder cancer. In non-invasive bladder cancers, a significantly greater extent of *HOXB2*, *FRZB*, and *KRT13* methylation was detected in grade 3 tumors as compared to grade 1 and 2 tumors. The methylation of these three genes was associated with aggressive non-invasive bladder cancer, with a significant 7.4-fold increased risk of developing high grade non-invasive bladder cancer. *HOXB2*, *FRZB*, and *KRT13* methylation status may be potential prognostic biomarkers to predict the likelihood of getting high grade non-invasive bladder cancer, that can subsequently be used to predict tumor recurrence or progression to muscle-invasion [46]. In luminal infiltrated and luminal papillary subtypes of bladder cancer, *HOXB2* mRNA expression was significantly upregulated. The knockdown of *HOXB2* reduced the cell proliferation, adhesion, and invasion of bladder cancer in vitro. Meanwhile, the overexpression of *HOXB2* induced cell proliferation and adhesion. These findings indicate that *HOXB2* promotes cell proliferation and adhesion in bladder cancer [47].

#### 3.1.6. *HOXB5*

*HOXB5* mRNA was found to be overexpressed in 70% of bladder cancer tissues. In cell lines, *HOXB5* mRNA expression was higher in most bladder cancer cell lines (5637, HT-1376, J82, RT4, T24, and TCCSUP) as compared to normal bladder cell lines. These findings suggest that *HOXB5* may function as an oncogene in bladder cancer. The proliferation and migration abilities of bladder cancer cells were significantly decreased in a *HOXB5* siRNA-transfected group as compared to the negative and mock control groups. Moreover, the *HOXB5* siRNA-transfected group suppressed clonogenicity in vitro, which further supports the oncogenic function of the *HOXB5* in bladder cancer [48].

In the same study, both the mRNA and protein levels of *HOXB5* were downregulated in a miR-7-transfected group as compared to that of the control group, thus indicating that *HOXB5* expression may be regulated by miR-7. This may be due to the presence of a SNP 1010A/G located within the miR-7 binding site in the 3′ untranslated region (UTR) of the *HOXB5*. An association was found between the frequency of the 1010A/G genotype and bladder cancer. The frequency of the 1010G genotype was higher in the bladder cancer group as compared to that of the normal control. Moreover, the frequency of 1010G genotype was also associated with high grade (grades 2 and 3) and high stage (stages T2–T4) tumors. These findings suggest that the presence of a SNP 1010A/G affects *HOXB5* expression [48]. Furthermore, *HOXB*5 expression was strongly associated with tumor stage, tumor grade, and the overall survival [49]. Taken together, *HOXB5* is overexpressed in bladder cancer and may act as oncogene. Hence, *HOXB5* could be a useful prognostic biomarker for bladder cancer.

#### 3.1.7. *HOXC4*, *HOXC5*, *HOXC6*, *HOXC11*, and *HOXD11*

Four genes (*HOXC4*, *HOXC5*, *HOXC6*, and *HOXC11*) located at the HOX C locus as well as one gene (*HOXD11*) located at the HOX D locus were not expressed in normal bladder tissues but expressed in most bladder cancer tissues. The expression of *HOXC4*, *HOXC5*, *HOXC6*, and *HOXC11* was detected in 60%, 86%, 100%, and 80% of bladder cancer tissues, respectively. In addition, *HOXD11* expression was detected in 57% of bladder cancer tissues. The findings suggest that these HOX genes, especially those located in the HOX C locus are involved in bladder tumorigenesis [50].

#### 3.1.8. *HOXC8*

Circular RNAs have been identified as crucial regulators of gene expression and may play important roles in cancer development. CircNT5E was found to promote bladder cancer cell proliferation and migration in vitro. CircNT5E acts as an miRNA sponge for miR-502-5p. The increased expression of miR-502-5p significantly inhibited both the mRNA and protein expression of *HOXC8*. The suppression of circNT5E inhibited both the mRNA and protein expression of *HOXC8*. The overexpression of *HOXC8* reversed the proliferation and migration inhibition of bladder cancer cells induced by the suppression of circNT5E expression. These findings suggest that the circNT5E/miR-502-5p/*HOXC8* pathway is a potentially important pathway associated with the development and progression of bladder cancer [51].

#### 3.1.9. *HOXA9*

The methylation status of a 4-gene panel (*HOXA9*, *EOMES*, *POU4F2*, and *ZNF154*) was significantly different between urine samples from bladder cancer patients and healthy individuals. The 84% sensitivity rate of the 4-gene panel methylation assay was higher than that of urinary cytology. These findings suggest that the methylation of the 4-gene panel can be used as a urinary biomarker for the early detection of bladder cancer [52]. In addition, a 6-gene panel (*HOXA9*, *EOMES*, *POU4F2*, *TWIST1*, *VIM*, and *ZNF154*) was highly hypermethylated in the urine of bladder cancer patients as compared to healthy individuals. The hypermethylation of the 6-gene panel had an 82–89% sensitivity in diagnosing bladder cancer and an 88–94% sensitivity in detecting bladder cancer recurrence. These findings suggest that methylation of the 6-gene panel can be utilized as a biomarker for the diagnosis and recurrence surveillance of bladder cancer [53].

In urine samples, the methylation status of a urinary biomarker panel (*HOXA9*, *ONECUT2*, *PCDH17*, *PENK*, *TWIST1*, *VIM*, and *ZNF154*) showed great prediction accuracy in predicting bladder cancer in patients with hematuria [54]. Another urinary biomarker panel (*HOXA9*, *ONECUT2*, *PCDH17*, and *POU4F2*) had a high positive predictive value and a negative predictive value in patients with hematuria [55]. These findings are useful for the detection of bladder cancer in patients with hematuria, which may help to reduce the need for cystoscopy in low-risk individuals. In relation to chemoresistance, nine genes (*HOXA9*, *ADD1*, *DBNDD2*, *EPAS1*, *GCNT4*, *RAPGEF5*, *TLR4*, *TSTD1*, and *ZNF582*) were found to be differentially expressed in cisplatin-resistant bladder cancer cell lines as compared to those that were cisplatin-sensitive. *HOXA9* promoter methylation was associated with greater resistance to cisplatin-based chemotherapy. These findings suggest that *HOXA9* promoter methylation could serve as a potential biomarker to predict chemoresistance in patients receiving cisplatin-based chemotherapy [56].

In high grade MIBC, a panel of hypermethylated genes (*HOXA9*, *CSPG2*, *HOXA11*, *HS3ST2*, *SOX1*, and *TWIST1*) was associated with muscle invasiveness. In addition, another panel of hypermethylated genes (*HOXA9*, *APC*, *CSPG2*, *EPHA5*, *EYA4*, *IPF1*, *ISL1*, *JAK3*, *PITX2*, *SOX1*, and *TWIST1*) predicted cancer-specific survival. These hypermethylated genes could potentially serve as biomarkers for the prognosis of bladder cancer [57]. The hypermethylation of a 3-gene panel (*HOXA9*, *ALDH1A3,* and *ISL1*) was observed in patients with NMIBC as compared to normal control group. The hypermethylation of the 3-gene panel is a predictor of bladder cancer recurrence, thus suggesting that the 3-gene panel is a promising biomarker for the early detection, diagnosis, and recurrence surveillance of bladder cancer [58]. There was significantly higher mean methylation of *HOXA9* and *ISL1* in recurrent and progressed high-grade NMIBC as compared to their non-recurrent counterparts. The concurrent methylation of *HOXA9* and *ISL1* predicted tumor recurrence and progression within one year, with a specificity of 90.9% and a positive predictive value of 91.7%. In relation to disease-specific mortality, the methylation of *HOXA9* and *ISL1* had an negative predictive value of 70.6% and 60%, respectively at a specificity of 57.1% [59]. These findings suggest that the methylation of *HOXA9* and *ISL1* have potential as prognostic biomarkers for bladder cancer.

#### 3.1.10. *HOXC9*

The miR-193a-3p promoted multi-chemoresistance of bladder cancer by repressing *HOXC9* expression. *HOXC9* is a direct target of miR-193a-3p in bladder cancer. Both the mRNA and protein expression of *HOXC9* were significantly higher in bladder cancer 5637 cells (multi-chemosensitive) as compared to resistant bladder cancer H-bc cells (multi-chemoresistant). The *HOXC9* influences the chemoresistance promoting effect of miR-193a-3p in bladder cancer cell lines and tumor-xenografted/nude mice through the regulation of the DNA damage response and oxidative stress pathways. These findings suggest that the identified genes involved in the miR-193a-3p/*HOXC9*/DNA damage response/oxidative stress pathway can serve as predictive of bladder cancer chemotherapy response [60].

#### 3.1.11. *HOXA10*

HOXA10 protein expression was significantly higher in bladder cancer tissues as compared to adjacent normal tissues. HOXA10 is overexpressed in bladder cancer tissues and contributes to the malignant behavior of bladder cancer cells. HOXA10 expression was significantly associated with the pathological grade and clinical stage of bladder cancer patients, thus suggesting that *HOXA10* may be involved in the pathological progression of bladder cancer. The knockdown of *HOXA10* in bladder cancer cells inhibited cell proliferation, migration, and invasion as well as decreased *MMP3* expression. Hence, *HOXA10* may be a potential biomarker for evaluating the progression of bladder cancer and a potential therapeutic target for bladder cancer [61].

*HOXA10* mRNA expression was found to be significantly upregulated in bladder cancer tissues and cell lines. Patients with high HOXA10 expression had lower survival rates as compared to those with low HOXA10 expression, thus suggesting that high HOXA10 expression is related to poorer bladder cancer prognosis. The knockdown of *HOXA10* inhibited cell migration and invasion in bladder cancer in vitro. In addition, *HOXA10* promoted the metastasis of bladder cancer by regulating *FOSLI*. These findings suggest that *HOXA10* may be an oncogene in bladder cancer that enhances the metastatic ability of bladder cancer cells [62].

#### 3.1.12. *HOXA11*

The hypermethylation of *HOXA11* was significantly more frequent in MIBC as compared to NMIBC. Thus, the silencing of *HOXA11* by hypermethylation could be used as a potential diagnostic and prognostic biomarker in bladder cancer [57].

#### 3.1.13. *HOXD10*

Ten miRNAs (miR-10b, miR-29a, miR-29b, miR-126, miR-142-5p, miR-146a, miR-146b-5p, miR-150, miR-155, and miR-342-3p) were upregulated, while three miRNAs (miR-143, miR-145, and miR-320) were downregulated in both primary and metastatic bladder cancers. The expression of *HOXD10* was detected in these samples. In comparison to primary bladder cancer, *HOXD10* was downregulated in metastatic bladder cancers. These findings suggest that *HOXD10* dysregulation may associated with bladder cancer metastasis [63].

Notably, miR-10b was significantly upregulated in metastatic bladder cancer cell lines and tissues. The expression of miR-10b promoted bladder cancer cell migration and invasion in vitro but suppressed metastasis in vivo. *HOXD10* and *KLF4* were identified as direct targets of miR-10b in bladder cancer. The overexpression of miR-10b decreased the protein expression of *HOXD10* and *KLF4*, whereas the downregulation of miR-10b increased the protein expression of *HOXD10* and *KLF4*. *MMP14* and E-cadherin may be the downstream targets of *HOXD10* and *KLF4* in the miR-10b-mediated suppression of bladder cancer metastasis. These findings indicate that miR-10b functions as a pro-metastatic miRNA in bladder cancer by targeting *HOXD10* and *KLF4*. Therefore, the inhibition of the miR-10b/*HOXD10*/*MMP14* and miR-10b/*KLF4*/E-cadherin axes may offer potential therapeutic targets for metastatic bladder cancer [64].

Propofol was found to significantly increase both the mRNA and protein expression of *HOXD10* in bladder cancer T24 cells. Propofol is a commonly used intravenous anaesthetic drug that exhibits antitumor properties in human cancers. It was reported that propofol inhibited bladder cancer cell viability, migration, and invasion in vitro. *HOXD10* is a direct target of miR-10b in bladder cancer cells. The overexpression of miR-10b in propofol-treated T24 cells significantly downregulated the expression of *HOXD10*. These findings suggest that propofol has a tumor-suppressive role in the regulation of cell viability, migration, and invasion of bladder cancer cells by targeting the miR-10b/*HOXD10* signaling pathway [65].

#### 3.1.14. *HOXA13*

*HOXA13* encodes a transcription factor that is involved in the differentiation and morphogenesis of genitourinary tracts [66]. HOXA13 was significantly higher expressed in bladder cancer tissues as compared to adjacent normal tissues. In addition, HOXA13 expression was significantly associated with lymphatic metastasis, higher tumor stage, and higher tumor grade. Bladder cancer patients with high HOXA13 expression had shorter overall survival and disease-free survival as compared to those with low HOXA13 expression. Moreover, HOXA13 was an independent prognostic factor for the overall survival of bladder cancer patients, in which increased HOXA13 expression level was associated with a poorer bladder cancer prognosis [13].

In the urine samples of bladder cancer patients, the mRNA expression of a panel of genes (*HOXA13*, *CDC2, IGFBP5*, and *MDK*) was detected in 48% of stage Ta tumors, 90% of stage T1 tumors, and 100% of stage >T1 tumors at a specificity of 85%. At a high specificity of >90%, a combination of *HOXA13* and *IGFBP5* had greater sensitivity in detecting low stage or grade tumors as compared to that of *MDK* and *CDC2*. In addition, the *HOXA13* and *CDC2* combination successfully distinguished grades 1–2 from grade 3 tumors and stage Ta from stage ≥T1 tumors, at a specificity and sensitivity of 80% [67].

The upregulation of *HOXA13*, *BLCA-4*, *IGF-1*, and *hTERT* was greater in stage Ta and T1 tumors as compared to stage >T1 tumors, with the greatest differential expression in *HOXA13* and *BLCA-4*. A combination of *HOXA13* and *BLCA-4* had an 80% specificity and sensitivity in differentiating low grade tumors from high grade tumors. At a specificity of 85%, a panel of biomarkers (*HOXA13*, *BLCA-4*, *hTERT*, and *IGF-1*) had detection rates of 48%, 90%, and 100% for Ta, T1, and >T1 stage tumors, respectively. Moreover, the panel had higher sensitivity than urine cytology across all tumor stages and grades. In addition, the panel detected 90% of stage T1 tumors as compared to 40% by cytology [68]. Based on pathway analysis, the genes in the biomarker panel play important roles in cell proliferation, differentiation, adhesion, and tumorigenesis [69]. Regardless of either being employed alone or in combination with other genes, *HOXA13* is a useful biomarker for stratifying bladder cancer patients into different groups of tumor stages and grades. Therefore, *HOXA13* helps to identify bladder cancer patient groups that require immediate surveillance and follow-up.

#### 3.1.15. *HOXB13*

IHC results show that the HOXB13 protein was heterogeneously expressed in bladder cancer tissues but was lowly expressed in normal bladder tissues. In addition, MIBC had a significantly higher HOXB13 protein expression as compared to NMIBC. The findings were corroborated by quantitative real time PCR (qPCR) results, in which a low or moderate increase (2–8-fold) of *HOXB13* mRNA expression was observed in all NMIBCs. In contrast, a significant increase (10–100-fold) of the *HOXB13* mRNA expression was observed in the majority of MIBCs. The findings indicate that HOXB13 expression is able to distinguish between NMIBC and MIBC phenotypes. NMIBCs had low nuclear and cytoplasmic HOXB13 expression, in which the loss of nuclear HOXB13 expression was significantly correlated with disease-free survival. The identification of nuclear HOXB13 expression in NMIBC contributed to better stratification of bladder cancer patients in relation to the risk of tumor recurrence. Interestingly, MIBC had low nuclear but high cytoplasmic HOXB13 expression. High cytoplasmic HOXB13 expression was significantly associated with MIBC. Taken together, the dysregulation and delocalization of HOXB13 suggest that HOXB13 plays a crucial role in tumor evolution and is a potential prognostic biomarker for bladder cancer [14].

#### 3.1.16. *MNX1*

The mRNA and protein expression of *MNX1* were markedly upregulated in bladder cancer cell lines as compared to primary normal urethral epithelial cells. In addition, *MNX1* expression was also significantly higher in bladder cancer tissues than in paired adjacent normal tissues. These findings show that *MNX1* is upregulated in bladder cancer. High MNX1 expression was significantly correlated with a shorter 5-year overall and relapse-free survival, thus suggesting that MNX1 expression is related to a poor bladder cancer prognosis. The overexpression of *MNX1* promoted cell proliferation in vitro and tumor growth in vivo, and promoted G1 to S phase transition of the cell cycle. In contrast, the downregulation of *MNX1* yielded the opposite effect. These findings suggest that *MNX1* promoted bladder cancer cell cycle progression, thus confirming that *MNX1* promotes bladder cancer cell proliferation. In addition, *MNX1* transcriptionally upregulated *CCNE1* and *CCNE2* by directly targeting their promoters. These findings suggest that *MNX1* may be an oncogene and a potential prognostic biomarker for bladder cancer [70].

#### 3.1.17. *BARX2*

The overexpression of *BARX2* significantly repressed the viability and invasion of bladder cancer cells. *BARX2* expression was upregulated by circSHPRH overexpression and downregulated by miR-942 overexpression. *BARX2* partially abrogated the tumorpromoting effects of circSHPRH knockdown on proliferation, migration, and invasion in bladder cancer cells. It was further confirmed that circSHPRH knockdown activated the Wnt/ß-catenin signaling pathway by regulating *BARX2*. These findings suggest that the circSHPRH/miR-942/*BARX2*/Wnt/ß-catenin axis might play an important role in bladder cancer progression and potentially serve as a therapeutic target for bladder cancer [71].

#### 3.1.18. *NANOG*

*NANOG* was significantly expressed in bladder cancer cell lines and tissues [72]. In bladder cancer tissues, differential expression patterns of stemness and EMT markers were found in NMIBC and MIBC. NANOG and SOX2 expression levels were significantly higher in MIBC as compared to NMIBC. In contrast, E-cadherin expression was significantly lower in MIBC than in NMIBC. These findings indicate that EMT and cancer stemness are enhanced in MIBC. An inverse correlation was found between E-cadherin and NANOG/SOX2 expression in bladder cancer tissues. E-cadherin expression was significantly associated with lower TNM stage and tumor grade. In contrast, NANOG expression significantly associated with higher TNM stage, higher tumor grade, and higher Ki-67 LI. *SOX2* expression was significantly correlated with higher TNM stage, higher tumor grade, and higher Ki-67 LI. These findings suggest that bladder cancer progression involves a decrease in E-cadherin and an increase in NANOG/SOX2 protein expression [73].

NANOG and BMI1 protein were highly expressed in the bladder cancer tissues. The overexpression of NANOG and BMI1 were correlated with a high tumor grade [74]. Positive NANOG expression was detected in all the formalin-fixed paraffin-embedded (FFPE) bladder cancer tissues with cytoplasmic, nuclear, and nuclear membrane localizations. The NANOG expression was increased across different tumor stages and grades, and was significantly associated with tumor invasion [75]. The downregulation of *NANOG* decreased bladder cancer cell migration and invasion as well as *MMP2* and *MMP9* mRNA levels. These findings suggest that the transcriptional activity of *NANOG* might be related to bladder cancer cell metastasis in vitro and that it has an influence on *MMP2* and *MMP9* expression [76].

The overexpression of *NANOG* remarkably increased the number and size of the spheres in bladder cancer UMUC3 and T24 cells, thus suggesting that *NANOG* promotes the self-renewal of bladder cancer cells. On the other hand, the overexpression of *WDR5* increased the expression of *NANOG*. These findings suggest that *WDR5* promotes bladder cancer cell self-renewal by mediating *NANOG* [77]. In another study, a significant correlation was found between *NANOG* and *CD44* expression. Bladder cancer patients with a high *NANOG* and *CD44* expression had poorer recurrence-free survival and overall survival rates. A combined *NANOG* and *CD44* expression were identified as independent prognostic biomarkers for recurrence-free survival and overall survival in bladder cancer [78].

Both the mRNA and protein expression of *NANOG*, *BMI1,* and *OCT4* (embryonic stem cell markers) were higher in cisplatin-resistant T24 cells (DR-T24) than that of parental T24 cells. In addition, the DR-T24 cells had a higher protein expression of CD44 (cancer stem cells marker) as compared to the parental T24 cells. This provides the first evidence for the presence of self-renewing populations of cancer stem cells (CSCs) in the DR-T24 and T24 cells. The CSCs could be enriched from T24 cells by cisplatin selection. These findings suggest that CSCs may be involved in the progression and cisplatin resistance of bladder cancer [74]. In bladder cancer T24 and 5637 cells, significantly higher mRNA and protein expression of *NANOG* and *OCT4* were observed after exposure to the highest concentration of mitomycin C (chemotherapy drug). These findings suggest that chemotherapy sorting might be a feasible method for isolating CSCs in bladder cancer [79]. In NMIBC, the ABCG2^hi^ side population cells had higher *NANOG*, *NOTCH 1*, and *SOX2* (embryonic stem cell markers) expression levels as well as increased colony forming efficiency when compared to the ABCG2^low^ non-side population cells [80]. Taken together, these findings suggest that NANOG might be useful as a potential biomarker for the diagnosis and prognosis of bladder cancer as well as for the identification of bladder CSCs.

#### 3.1.19. *NANOGP8*

The mRNA expression of *NANOGP8* was detected in bladder cancer tissues. The percentage of *NANOGP8*-transfected cells at the S phase of cell cycle was higher than that of the control cells. These findings indicate that *NANOGP8* promotes cells to enter into the S phase of cell cycle. In addition, *NANOGP8* also promotes cell proliferation, in which a significantly increase in cell proliferation was observed in *NANOGP8*-transfected cells as compared to the control cells. These findings suggest that *NANOGP8* may play an important role in bladder tumorigenesis [81].

#### 3.1.20. *NKX2-8*

Both the mRNA and protein expression of *NKX2-8* were markedly reduced in bladder cancer tissues when compared to normal bladder tissues. Moreover, patients with negative NKX2-8 expression had a higher tumor recurrence risk as compared to those with positive NKX2-8 expression. The overexpression of *NKX2-8* in bladder cancer T24 cells significantly inhibited cell proliferation in vitro and in vivo, whereas the silencing of *Nkx2-8* in bladder cancer 5637 cells dramatically enhanced cell proliferation. The silencing of *NKX2-8* also resulted in an acceleration of the G1/S transition, downregulation of p27^kip1^, upregulation of cyclin D1 and p-FOXO3a, and an increase in MEK/ERK pathway activity. Meanwhile, the overexpression of *NKX2-8* yielded the opposite effect. These findings indicate that *NKX2-8* regulates cyclin D1 and p27^kip1^ through the MEK/ERK pathway in bladder cancer cells. *Nkx2-8* potentially acts as a tumor suppressor gene in bladder cancer progression and could be developed as prognostic biomarker of bladder cancer [82].

*NKX2-8* inhibited the EMT phenotype in bladder cancer cells. The knockdown of *NKX2-8* promoted the invasion and metastatic potential of bladder cancer cells, whereas the overexpression of *NKX2-8* inhibited this potential. These findings suggest that *NKX2-8* acts as a negative regulator of aggressive metastasis of bladder cancer. In addition, *NKX2-8* downregulated *TWIST1* expression, whereby *NKX2-8* bound to the *TWIST*1 promoter locus and transcriptionally repressed *TWIST1*. The upregulation of *TWIST1* reversed EMT inhibition by *NKX2-8* as well as restored the invasive phenotype of bladder cancer cells. In bladder cancer tissues, *NKX2-8* expression was inversely correlated with *TWIST1* expression. Bladder cancer patients with tumors that were *NKX2-8* positive and low *TWIST1* expression had better prognosis as compared to those that were negative *NKX2-8* and harbored high *TWIST1* expression. These findings suggest that *NKX2-8* inhibited EMT in bladder cancer through the transcriptional repression of *TWIST*1. In addition, the *NKX2-8*/*TWIST1* axis plays a crucial role in bladder cancer EMT and may be a potential therapeutic target for bladder cancer [83].

#### 3.1.21. *NKX6-2*

Eight genes (*NKX6-2*, *A2BP1*, *CA10*, *DBC1, MYO3A, NPTX2*, *PENK*, and *SOX11*) were significantly highly methylated in the urine sediments of bladder cancer patients as compared to that of control subjects. A 5-gene panel was identified (*NKX6-2*, *CA10*, *DBC1*, *MYO3A*, and *PENK* or *SOX11*) and the panel had an 85% sensitivity and a 95% specificity for the detection of bladder cancer. The detection rates of bladder cancer in NMIBC and MIBC were 81% and 90%, respectively. In addition, the detection rate was 85% in both primary and recurrent bladder cancers [84]. In recurrent and progressed bladder tumors, the frequency of *NKX6-2* methylation was higher than that of non-recurrence bladder tumors [59]. Taken together, these findings suggest that *NKX6-2* methylation is a potential biomarker for the detection of bladder cancer that if introduced, can potentially reduce the frequency of using cystoscopy for bladder cancer surveillance. In addition, *NKX6-2* methylation also potentially predicts the risk of recurrence and the progression of bladder cancer.

#### 3.1.22. *TLX1*

*BCAR4* promoted the cell proliferation, migration, and invasion of bladder cancer. *BCAR4* can directly bind to miR-644a and regulate *TLX1* expression. The knockdown of *BCAR4* or the overexpression of miR-644a significantly decreased the expression of *TLX1*. On the other hand, the overexpression of *TLX1* or the inhibition of miR-644a increased bladder cancer cell migration. This is the first study to show that *BCAR4* and miR-644a can regulate the expression of *TLX1* and the expression of *TLX1* was associated with bladder cancer progression [85].

#### 3.1.23. *TLX3*

*TLX3* was methylated in cisplatin-resistant cells (T24DDP10 and KK47DDP20) and bladder cancer tissues in which *TLX3* mRNA expression was not detected. In contrast, *TLX3* was unmethylated in cisplatin-sensitive cells (T24 and KK47) and non-malignant bladder tissues in which *TLX3* mRNA expression was detected. These findings indicated that *TLX3* may be involved in cell proliferation and cisplatin resistance in bladder cancer. The overexpression of the TLX3 protein in T24DDP10 cells increased cell proliferation and restored cisplatin sensitivity. In contrast, the knockdown of *TLX3* in T24 cells inhibited cell proliferation and increased cisplatin resistance. These findings suggest that *TLX3* expression induced bladder cancer cell proliferation and cisplatin resistance, hence increasing cisplatin toxicity. In contrast, the loss of *TLX3* expression resulted in cisplatin resistance in bladder cancer. Bladder cancer patients with *TLX3* hypermethylation had higher succinate dehydrogenase (SD) activity (decreased chemosensitivity to cisplatin). Patients with *TLX3* unmethylation had lower SD activity (increased chemosensitivity to cisplatin). These findings indicate that *TLX3* methylation in bladder cancer is associated with cisplatin resistance. Hence, *TLX3* methylation can be utilized as a predictive biomarker for cisplatin resistance in bladder cancer [86].

#### 3.1.24. *VAX1*

The methylation frequency of eight genes (*VAX1*, *ECEL1*, *KCNV1*, *LMX1A*, *PROX1*, *SLC6A20, TAL1*, and *TMEM26*) was significantly higher in the urine of bladder cancer patients as compared to that of normal controls. A 5-gene panel (*VAX1*, *CFTR*, *KCNV1*, *PROX1*, and *TAL1*) had an 88.7% sensitivity and an 87.3% specificity for the diagnosis of bladder cancer. Moreover, the 5-gene panel had an accuracy of 81.3%, which is comparable to that of cystoscopy. These findings suggest that the 5-gene panel is a promising diagnostic biomarker for the early non-invasive detection and surveillance of bladder cancer. Remarkably, *VAX1* and *LMX1A* methylation were more frequently detected in urine samples from recurrent bladder cancer patients as compared to that of newly diagnosed bladder cancer patients, thus suggesting that the methylation of *VAX1* and *LMX1A* was associated with bladder cancer recurrence. These findings suggest that *VAX1* and *LMX1A* hypermethylation may be useful as a biomarker for predicting bladder cancer recurrence [87].

#### 3.1.25. *VAX2*

In NMIBC, *VAX2* had significantly higher methylation frequency in high-grade tumors as compared to low or intermediate-grade tumors. In addition, the promoter methylation of *VAX2* was significantly associated with reduced gene expression. The findings suggest that *VAX2* may be a potential therapeutic target for bladder cancer [88].

### 3.2. The CERS Homeobox Class

####  *CERS2*

The protein expression of CERS2 was detected in bladder cancer tissues across different stages, in which it was detected in 90.5%, 62.5%, and 33.3% of stages I, II, and III tumors, respectively. Bladder cancer patients with negative CERS2 expression had significantly poorer survival as compared to those with positive CERS2 expression. In addition, the differential mRNA expression of *CERS2* was detected in freshly frozen bladder cancer tissues, paraneoplastic, and normal bladder tissues. Both the protein and mRNA expression of *CERS2* were downregulated in advanced bladder cancer tissues and they were significantly associated with tumor stage, the depth of tumor invasion, and tumor recurrence. The loss of *CERS2* expression was strongly associated with progression and invasion of bladder cancer. These findings suggest that *CERS2* serves as a tumor metastasis suppressor gene and may serve as a prognostic biomarker for bladder cancer [74]. The potential utility of *CERS2* as a predictive biomarker of response to oncolytic virotherapy has also been investigated [89]. Unfortunately, the potential of *CERS2* as a predictive biomarker was found to be limited but may have the potential as a therapeutic target for bladder cancer, since the siRNA-mediated downregulation of *CERS2* expression resulted in reduced bladder cancer cell migratory potential [90].

*CERS2* (also known as *LASS2*) mRNA expression was significantly different across four bladder cancer cell lines (EJ, EJ-M3, T24, and BIU-87 cells). The invasiveness of EJ, T24, and BIU-87 cells was not significantly different. However, the invasiveness of EJ-M3 cells was significantly different from that of EJ, T24, and BIU-87 cells. Notably, EJ-M3 was the most aggressive among these cell lines. In addition, *LASS2* mRNA expression was significantly correlated with diverse cell proliferation, invasion, and metastasis. At the protein level, a lower LASS2 protein expression level was observed in the more aggressive cell line. These findings suggest that *LASS2* acts as a metastasis suppressor gene in bladder cancer and potentially serves as a biomarker for the prognosis of bladder cancer [89].

Apart from mRNA expression, LASS2 protein expression was also significantly associated with tumor stage, the depth of tumor invasion, and tumor recurrence. The loss of LASS2 protein expression was strongly associated with the progression and invasion of bladder cancer. Patients with LASS2-negative bladder cancer had significantly poorer survival than those with LASS2-positive bladder cancer. Taken together, these findings support the role of LASS2 as a metastasis suppressor gene in bladder cancer and its potential utility as a prognostic biomarker for bladder cancer [91].

It was reported that *LASS2* was a target of miR-20a, whereby miR-20a negatively regulate *LASS2* gene at both the mRNA and protein levels. Increasing the miR-20a expression level was closely related to aggressive clinicopathological parameters. Expression of miRNA-20a promotes bladder cancer cell proliferation, invasiveness, and migration by targeting *LASS2* [92]. In advanced bladder cancers, the expression of LASS2 was found to be downregulated. The lower expression of LASS2 was associated with a higher tumor stage and poorer survival as compared to tumors with normal expression of LASS2. Thus, LASS2 might be a potential biomarker of bladder cancer progression. In addition, hsa-miR-3622a was found to be negatively correlated with *LASS2*. The overexpression of miR-3622a promoted bladder cancer cell proliferation and invasion in vitro by downregulating *LASS2* [93].

The expression of miR-9 was higher in bladder cancer tissues as compared to normal tissues. It was found that miR-9 increased cell proliferation, invasion, cell cycle progression, and chemoresistance. *LASS2* is also a direct target of miR-9 in bladder cancer and the transfection of a miR-9 mimic was found to downregulate the expression of *LASS2*. These findings indicate that miR-9 upregulation was associated with a malignant phenotype of bladder cancer and promoted cell proliferation and chemoresistance by targeting *LASS2* in bladder cancer [94]. Another miRNA expression, miR-98, was higher in bladder cancer tissues and cell lines as compared to adjacent normal tissues and SV-HUC-1 cells. The miR-98 mimic was found to promote cell proliferation, inhibit apoptosis, and increase cisplatin/doxorubicin resistance in bladder cancer T24 cells. Meanwhile, the miR-98 inhibitor inhibited cell proliferation, promoted apoptosis, decreased chemoresistance in bladder cancer BIU-87 cells. In addition, miR-98 could regulate mitochondrial fission/fusion balance and mitochondrial membrane potential. *LASS2* is also a direct target of miR-98 in bladder cancer and a negative correlation was found between *LASS2* and miR-98 mRNA levels. The overexpression of *LASS2* induced mitochondrial fusion and downregulated mitochondrial potential. Meanwhile, *LASS2* siRNA abrogated the chemoresistance effects of the miR-98 mimic. These findings demonstrate that miR-98 promotes bladder cancer chemoresistance and regulates mitochondrial function by targeting *LASS2* [95].

### 3.3. The CUT Homeobox Class

#### 3.3.1. *CUX1*

The *PIK3CA* is overexpressed in bladder cancer and its expression is regulated by *CUX1*. *PIK3CA* promotes bladder cancer cell proliferation, migration, invasion, and angiogenesis in vitro. In addition, *PIK3CA* also enhances the growth and metastasis of bladder cancer in vivo. The overexpression of *CUX1* elevates both the mRNA and protein expression of *PIK3CA*, whereas the knockdown of *CUX1* reduces *PIK3CA* expression. *CUX1* stimulates the transcription activity of *PIK3CA* in bladder cancer cells via direct interaction with the binding site of the *PIK3CA* promoter. *CUX1* upregulates *PIK3CA* expression in bladder cancer and this in turn, further activates EMT. These findings indicate that *PIK3CA* is targeted by *CUX1* and the activation of the *CUX1*/*PIK3CA* axis activates the EMT pathway, which may contribute to the promotion of invasion and metastasis in bladder cancer [96].

#### 3.3.2. *ONECUT2*

A combination of five methylation markers *(ONECUT2, MEIS1, OSR1, OTX1,* and *SIM2)* resulted in a bladder cancer prediction model, with a sensitivity of 85% and a specificity of 87% [97]. Another combination of three methylation markers *(ONECUT2, OTX1*, and *TWIST1)*, together with the mutation status of *FGFR3*, *HRAS,* and *TERT*, and patient age were included in another prediction model that had a 97% sensitivity and an 83% specificity. Based on these prediction models, epigenetic profiling using urine samples of patients with hematuria may reduce the need for invasive cystoscopy in the management of low-risk patients [98,99].

#### 3.3.3. *SATB1*

SATB1 expression was increased significantly in bladder cancer cell lines as compared to normal bladder cell lines. The knockdown of *SATB1* in two high-grade bladder cancer cell lines (TCCSUP and 5637) showed opposite functional roles. Compared to the non-silencing control, the *SATB1*-knockdown TCCSUP cells showed a decrease in cell proliferation and an increase in sensitivity to cisplatin. In contrast, *SATB1*-knockdown 5637 cells showed an increase in cell proliferation and resistance to cisplatin. The differences in cisplatin resistance and proliferation between the *SATB1*-knockdown TCCSUP and 5637 cells may be due to differential gene expressions of *SATB1* (and other modifying genes) between the cell lines [100].

Both the mRNA and protein expression of *SATB1* and *ERBB2* were significantly upregulated in bladder cancer tissues when compared to normal bladder tissues. A positive correlation was found between *SATB1* and *ERBB2*. Elevated mRNA and protein levels of both genes were significantly associated with tumor stage and lymph node involvement. These findings suggest that *SATB1* and *ERBB2* may serve as potential biomarkers for predicting the aggressive behavior and poor prognosis of bladder cancer [101]. In NMIBC tissues, both the mRNA and protein expression of *SATB1* was significantly upregulated as compared to that of adjacent normal tissues. The SATB1 expression was remarkably higher in MIBC tissues than in NMIBC tissues. The same trend was observed in which the SATB1 expression level was higher in the metastatic bladder cancer T24 cells than in the non-metastatic bladder cancer BIU-87 cells. These findings suggest that SATB1 expression is associated with aggressive tumor phenotypes. Moreover, a positive correlation was found between SATB1 and EMT markers as well as vimentin. The overexpression of *SATB1* in the BIU-87 cells significantly increased cell migration and invasiveness, and also caused changes in the cell morphology; from cobblestone-like morphology (epithelial phenotype) to spindle-like fibroblastic morphology (mesenchymal phenotype), thus indicating an induction of EMT. In contrast, the downregulation of *SATB1* yielded the opposite effect. A significant correlation was found between a higher *SATB1* expression and shorter survival times. In addition, *SATB1* expression was to be a significant and independent prognostic factor for bladder cancer [102].

SATB1 expression was detected in 38.8% of FFPE bladder cancer tissues but was not expressed in normal bladder tissues. Higher SATB1 expression was found in stage T2–T4 tumors when compared to stage T1 tumors, thus suggesting that SATB1 plays an important role in bladder cancer progression. SATB1 was also highly expressed in bladder cancer 5637 and T24 cells but was not expressed in the SV-HUC-1 cells. The downregulation of *SATB1* expression decreased cell proliferation and increased cisplatin-induced apoptosis. *SATB1* downregulation also decreased both the mRNA and protein expression of cyclin D1 (cell proliferation factor) and cyclin E (cyclin D inhibitor) but increased the expression of cleaved caspase 3 (an apoptosis-related gene). These findings suggest that *SATB1* is overexpressed in bladder cancer and *SATB1* also regulates cell proliferation and cisplatin-induced apoptosis by modulating cyclin D1, cyclin E, and caspase 3. Therefore, *SATB1* may be a therapeutic target for bladder cancer [103].

### 3.4. The LIM Homeobox Classes

####  *ISL1* and *LHX5*

The positive expression of ISL1 and LHX5 was detected in 94% and 98% of FFPE bladder cancer tissues, respectively. The proportion of tumors expressing high ISL1 and LHX5 levels were highest in late-stage and high-grade tumors, respectively. In addition, a significant association was found between LHX5 expression and tumor grade. These findings suggest that *ISL1* and *LHX5* play an important role in bladder tumorigenesis [104].

### 3.5. The POU Homeobox Class

####  *POU5F1*

The expression of *POU5F1* was detected in bladder cancer tissues and cell lines (J82, T24, TCCSUP, and TSGH-8301) but was not detected in normal bladder tissues and SV-HUC-1 cell line [105,106]. These findings were in agreement with those studies that reported that positive POU5F1 protein expression was detected in 68%–81.6% of bladder cancer tissues. In contrast, normal bladder tissues showed negative POU5F1 expression [107,108]. Taken together, the presence of POU5F1 expression in bladder cancer tissues and cell lines suggests that POU5F1 promotes bladder tumorigenesis [105,106,107,108]. In addition, POU5F1 positive expression was associated with high-grade (grades 2 and 3) tumors when compared to low-grade (grade 1) tumors, and was also significantly associated with higher recurrence incidence [107]. Intense POU5F1 expression was associated with disease progression, greater metastasis, and shorter cancer-related survival as compared to tumors with low or moderate POU5F1 expression [105]. These findings suggest that POU5F1 may play a role in the development of high-grade tumors and in the recurrence in bladder cancer. Hence, POU5F1 could be useful as a diagnostic biomarker as well as a prognosis biomarker for the prediction of recurrence risk and the survival of bladder cancer patients [105,107,109,110].

More rapid cell migration was observed in both human TCCSUP/*POU5F1* cells and murine MBT-2/*POU5F1* cells when compared to control cells. In addition, the overexpression of *POU5F1* in TCCSUP/*POU5F1* cells and MBT-2/*POU5F1* cells exhibited greatly enhanced cell migration and invasion. In contrast, POU5F1 knockdown reduced cell migration and invasion in both cells. These findings suggest that *POU5F1* is involved in bladder cancer cell migration and invasion in vitro. In animal models, mice inoculated with MBT-2/*POU5F1* cells had a higher frequency of tumor nodules, more lung tumor lesions, and a greater number of visible metastatic pulmonary nodules as compared to those inoculated with control cells. The overexpression of *POU5F1* increased *FGF-4*, *MMP2*, and *MMP13* expression that cell displayed a metastatic phenotype. In contrast, the knockdown of *POU5F1* yielded the opposite effect. These findings suggest that *POU5F1* overexpression in bladder cancer upregulates *FGF-4*, *MMP2*, and *MMP13* expression, hence promoting metastasis of bladder cancer in vivo. In a syngeneic murine tumor model, mice bearing *MBT-2* tumors treated with an oncolytic adenovirus, Ad5WS4 (E1B-55 kD-deleted adenovirus driven by the *POU5F1* promoter), significantly suppressed tumor growth and prolonged survival as compared to the control group [105].

The majority of the FFPE bladder cancer tissues showed positive OCT4 (also known as *POU5F1*) expression with a variety of intensities, thus indicating that OCT4 is heterogeneously expressed in bladder cancer cells. The OCT4 protein was mainly localized in the nucleus of bladder cancer cells, with a low to moderate cytoplasmic localization. The low expression of OCT4 was found in 72% of cases, whereas the high expression of OCT4 was found in 28% of cases. A highly significant correlation was found between OCT4 expression intensity and tumor stage. In addition, a significant correlation was also found between OCT4 expression intensity and lamina propria/muscularis propria invasion. Taken together, OCT4 expression was associated with advanced tumor stage, the involvement of the lamina propria and muscularis propria invasion. These findings suggest that the expression of OCT4 promotes bladder tumor progression and aggressive bladder cancer cell phenotype. Hence, *OCT4* can be a potential prognostic biomarker and therapeutic target for bladder cancer [111].

The expression of embryonic stem cell markers (*OCT4* and *NANOG*) was significantly higher in bladder cancer T24 and 5637 cells as compared to the controls after exposure to the highest concentration of mitomycin C, thus indicating that *OCT4* and *NANOG* expression were higher in drug-resistant cells. These findings suggest that chemotherapy sorting might be a feasible method for identifying CSCs in bladder cancer [79].

The mRNA expression level of *OCT4* were significantly higher in bladder cancer tissues as compared to normal bladder tissues. Patients with OCT4 high-expressing tumors had significantly shorter recurrence-free interval than those with OCT4 low-expressing tumors. The expression level of OCT4 were significantly higher in recurrent tumors than those in primary tumors. In addition, OCT4 expression was dramatically increased in bladder cancer TCCSUP cells treated with anticancer drugs (i.e., cisplatin, doxorubicin, and 5-fluorouracil). The knockdown of *OCT4* expression increased the drug sensitivity of bladder cancer cells, whereas the overexpression of *OCT4* expression yielded the opposite effect. On the other hand, *CD44* was colocalized with OCT4, thus suggesting that the concurrent upregulation of *CD44* and *OCT4* during cisplatin treatment. *CD44*-positive bladder cancer cells that expressed OCT4 were dramatically increased in bladder cancer cell lines after cisplatin treatment. Furthermore, the colocalization of *CD44* with *OCT4* was detectable in human bladder tumor xenografts that were resistant to cisplatin treatment. These findings demonstrate that *CD44*-positive bladder cancer cells expressing *OCT4* are induced by cisplatin, which may contribute to drug resistance. Taken together, the overexpression of *OCT4* in bladder cancer confers resistance against cisplatin in vitro and in vivo, thus suggesting that the mechanism underlying acquired drug resistance may involve *OCT4* overexpression [112].

The protein expression level of OCT4 was found to be positively correlated with KPNA2 expression level in bladder cancer tissues. Upregulated OCT4 and KPNA2 expression positively correlated with tumor stage and pathological types. The high protein expression of OCT4 and KPNA2 were independent prognostic predictors of bladder cancer. Both *OCT4* and *KPNA2* were also found to be upregulated in bladder cancer J82 and T24 cells. The mRNA expression level of *OCT4* was downregulated when *KPNA2* was silenced. The silencing of *OCT4* and *KPNA2* decreased bladder cancer cell proliferation and migration while increasing apoptosis. These findings demonstrate that both *OCT4* and *KPNA2* promotes cell proliferation and migration as well as inhibit apoptosis in vitro. In addition, the knockdown of *KPNA2* inhibits the nuclear translocation of *OCT4*, thus suggesting that the process of OCT4 nuclear transportation in bladder cancer was regulated by *KPNA2* [113].

### 3.6. The PRD Homeobox Class

#### 3.6.1. *ALX4*

*ALX4* was methylated in 25% of the urine sediments of bladder cancer patients, whereas no methylation of *ALX4* was observed in non-malignant urinary lesions. Methylation profiling of an 11-gene panel (*ALX4*, *ABCC6*, *BRCA1*, *CDH13*, *CFTR*, *HPR1*, *MINT1*, *MT1A*, *RASSF1A*, *RPRM*, and *SALL3*) had a 91.7% sensitivity and an 87% specificity for the detection of bladder cancer. Based on the methylation profiles of the 11-gene panel, >75% of stage 0a and 88% of stage 1 tumors were detected. The methylation profiling of the 11-gene panel may contribute to the development of diagnostic biomarkers for the early detection of bladder cancer [114].

#### 3.6.2. *DUXAP10*

The expression of *DUXAP10* in bladder cancer cell lines (5637, EJ, RT4, T24, TCCSUP, and UMUC3) was significantly higher as compared to that of SV-HUC-1 cell line, thus suggesting that *DUXAP10* is overexpressed in bladder cancer. The knockdown of *DUXAP10* was found to inhibit bladder cancer cell proliferation, induce cell cycle arrest at the G0/G1 phase, and promote apoptosis. These findings indicate that *DUXAP10* accelerates proliferation by promoting the G0/G1 to S phase transition and suppressing apoptosis in bladder cancer cells. In addition, the knockdown of *DUXAP10* also inhibited the PI3K/Akt/mTOR signaling pathway in bladder cancer cells. These findings suggest that *DUXAP10* inhibited bladder cancer cell proliferation and induced apoptosis through the PI3K/Akt/mTOR signaling pathway. Taken together, *DUXAP10* plays an important role in bladder cancer and the inhibition of *DUXAP10* is a potential therapeutic target for bladder cancer [115].

#### 3.6.3. *OTX1*

*OTX1* was overexpressed in bladder cancer tissues and cell lines, which was significantly associated with a poor prognosis of bladder cancer. *OTX1* silencing significantly reduced cell viability and inhibited cell proliferation in bladder cancer. In contrast, the overexpression of *OTX1* yielded the opposite effect. In addition, *OTX1* silencing induced cell cycle arrest in the G0/G1 phase and inhibited tumor growth in vivo. Taken together, these findings suggest that *OTX1* promotes cell proliferation and cell cycle progression in vitro as well as tumor growth in vivo. Hence, *OTX1* may promote bladder cancer progression and might be a potential prognostic biomarker for bladder cancer [116].

#### 3.6.4. *PAX2*

*PAX2* expression has been frequently detected in various human cancers, including brain, breast, colon, lung, and ovarian cancers. Moderate mRNA expression and the detectable protein expression of *PAX2* were observed in bladder cancer EJ cells. The knockdown of *PAX2* inhibited and induced apoptosis in the EJ cells as compared to the control cells, thus suggesting that *PAX2* is essential for the proliferation and survival of bladder cancer cells [117].

#### 3.6.5. *PAX5*

PAX5 protein expression is not frequently detected in bladder cancer. Nuclear PAX5 protein was only detected in 0.2–10% of bladder cancer cases, thus suggesting that *PAX5* may only play a minor functional role in bladder cancer [118,119]. However, *PAX5* mRNA expression has been detected in 79% of bladder cancer tissues and cell lines (RT112, HT-1376 and MGH-U1) [120,121,122]. A significantly higher *PAX5* mRNA expression was detected in pT2 and pT3 tumors when compared to pT1 tumors [89]. *PAX5* mRNA expression was detected in 83.3% of bladder cancer patients, whereas negative *PAX5* expression was detected in the control group [122]. Patients with a high *PAX5* expression had lower 3-year recurrence-free survival rate and 3-year progression-free survival rate as compared to those with negative *PAX5* expression [121]. These findings suggest that *PAX5* may in fact play a role in bladder cancer and potentially serve as a diagnostic biomarker for bladder cancer [120,121,122].

*PAX5* was significantly increased in cisplatin-resistant bladder cancer tissues and cell lines. The knockdown of *PAX5* improved cisplatin sensitivity of bladder cancer cells, whereas the overexpression of *PAX5* increased cisplatin resistance. *PTGS2* is the direct downstream transcriptional target of *PAX5*. These findings suggest that the dysregulation of the *PAX5*/*PTGS2* cascade plays a crucial role in the induction of cisplatin resistance in bladder cancer. Hence, gene silencing approaches targeting this pathway may provide a therapeutic strategy for overcoming cisplatin resistance in bladder cancer [123].

#### 3.6.6. *PAX6*

The methylation rate of the *PAX6* promoter was found to be significantly higher in bladder cancer tissues as compared to adjacent normal tissues. Interestingly, the methylation rate of the *PAX6* promoter was higher in the adjacent normal tissues from patients with pTa tumors as compared to those with pT1 tumors. The *PAX6* methylation level in non-malignant tissues was associated with tumor stages, thus suggesting that a putative field cancerization effect in the normal adjacent bladder mucosa. Hence, *PAX6* methylation has diagnostic implications for bladder cancer [124].

In high-grade pT1 tumors, the methylation profile of *PAX6*, *ATM*, *CHTR*, and *RB1* independently predicted the recurrence of bladder cancer. In addition, the methylation of *PAX6, ESR1*, *MGMT*, *PTEN*, *RB1*, and *TP73* had a 100% positive predictive value for progression. The methylation of *PAX6* had a 100% positive predictive value for progression and disease-specific survival. These findings suggest that methylation profiling of *PAX6* is a potential prognostic biomarker for identifying bladder cancer patients with a higher recurrence risk [125]. The hypermethylation of *PAX6* was found in T1G3 tumors, whereas no hypermethylation was found in the normal bladder tissues. T1G3 bladder cancer patients who treated with Bacillus Calmette–Guerin (BCG) immunotherapy had a higher recurrence rate when the tumors harbored *PAX6* methylation as compared to those that were unmethylated. Taken together, *PAX6* methylation status is able to distinguish bladder cancer patients who are most likely to have a high risk of recurrence [126].

#### 3.6.7. *PAX8*

Nested urothelial carcinoma is a rare histological variant of bladder cancer. A significant proportion of nested urothelial carcinoma cases express PAX8. The PAX8 immunoreactivity was strong in 30% and moderate in 26% of the cases. These findings suggest that positive PAX8 expression may be useful in avoiding the misdiagnosis of nested urothelial carcinoma as nephrogenic adenoma, particularly in cases with limited sampling [127]. In addition, positive PAX8 expression was detected in 52% and 67% of primary and metastatic nested urothelial carcinoma, respectively [128].

Nuclear PAX8 expression was detected in 93% of non-invasive urothelial tumors, including papillary urothelial neoplasm of low malignant potential (PUNLMP), non-invasive low-grade papillary urothelial carcinoma (NILGC), and non-invasive high-grade papillary urothelial carcinoma (NIHGC). In contrast, all the normal bladder tissues examined showed negative PAX8 expression. These findings suggest that *PAX8* may play a role in the neoplastic transformation of bladder cancer cells [129].

#### 3.6.8. *PRRX1*

Weighted gene co-expression network analysis (WGCNA), protein–protein interaction (PPI), the Gene Expression Omnibus (GEO) database, and survival analysis identified five hub genes related to prognosis, including *PRRX1*, *ACTA2*, *COL5A1*, *DCN*, and *LUM*. In addition, the expression of the five hub genes was closely related to tumor invasion, whereby it was statistically significant different in T2-T4 tumors. Thus, the expression of the five hub genes can be potentially used as biomarkers and for the targeted therapy of MIBC [130].

A total of 14 DEGs were found to be associated with the overall survival of bladder cancer. Of these, 10 immune-associated DEGs were demonstrated to be predictive of the prognosis in bladder cancer. Five genes (*PRRX1*, *BTBD16*, *OLFML2B*, *SPINK4*, and *SPON2)* have not been previously reported to be associated with the prognosis of bladder cancer. These findings suggest that the 10 genes that were closely related to prognosis probability may have potential as prognostic biomarkers for bladder cancer [131].

### 3.7. The PROS Homeobox Class

####  *PROX1*

The lncRNA LNMAT2 was found to upregulate *PROX1* expression by increasing H3K4 trimethylation level of the *PROX1* promoter. *PROX1* promotor hypermethylation was also found to be associated with lymphatic metastasis in bladder cancer [132].

### 3.8. The SINE Homeobox Class

####  *SIX4*

*SIX4* is a direct target of miR-203a in bladder cancer. The mRNA expression level of *SIX4* is negatively corrected with that of miR-203a. The miR-203a was found to be significantly inhibit bladder cancer cell proliferation, migration, and invasion. Nevertheless, miR-203a overexpression induce apoptosis and G2/M phase cell cycle arrest. The miRNA was found to suppress EMT progression and the PI3K/Akt signaling pathway in vitro. The downregulation of *SIX4* in bladder cancer cells inhibit cell migration and invasion, and induce G2/M phase cell cycle arrest and apoptosis. Meanwhile, the overexpression of *SIX4* diminished the effects of miR-203a on bladder cancer cells in vitro. These findings suggest that miR-203a acts as a tumor suppressor in bladder cancer by inhibiting the potentially oncogenic *SIX4*. The expression ratios of miR-203a/*SIX4* may be a useful as a prognostic biomarker for bladder cancer [133].

### 3.9. The TALE Homeobox Class

####  *TGIF1*

Higher cell migration, reactive oxygen species (ROS) production, *TGIF1* and p67^phox^ expression, and AKT^s473^ phosphorylation were detected in invasive bladder cancer T24 cells as compared to the non-invasive bladder cancer RT4 cells. The knockdown of *TGIF1* in the T24 cells significantly reduced cell migration and invasion, ROS production, and Nox2 and p67^phox^ expression. Meanwhile, the overexpression of *TGIF1* in the RT4 cells yielded the opposite effect. Hence, *TGIF1* was found to promote bladder cancer cell migration and invasion, induced the production of ROS, and regulated Nox2 and p67^phox^ expression. The PI3K/AKT pathway is involved in the production of *TGIF1*-induced ROS, while the activation of *TGIF1*-induced Nox2/p67^phox^ is associated with cell migration and invasion [134]. The overexpression of *TGIF1* was also found to enhance cell migration, activate p-AKT (Ser473), and reduce cellular sensitivity to gemcitabine. Meanwhile, the knockdown of *TGIF1* yielded the opposite effect. These findings suggest that *TGIF1* contributes to gemcitabine resistance of bladder cancer via AKT activation [135].

### 3.10. The ZF Homeobox Class

#### 3.10.1. *ADNP*

Both the mRNA and protein expression of *ADNP* were significantly higher in patients with progressive bladder cancer as compared to those with non-progressive bladder cancer. All the patients underwent TURBT and treated with intravesical chemotherapy. Thus, these findings suggest that *ADNP* overexpression was associated with bladder cancer progression in patients treated with intravesical chemotherapy. Bladder cancer patients with a high *ADNP* expression had significantly shorter tumor-free survival after chemotherapy. *ADNP* was a prognostic risk factor for bladder cancer progression after intravesical chemotherapy treatment. These findings support that the upregulation of ADNP in bladder cancer tissues was associated with poor prognosis in intravesical chemotherapy-treated patients. The knockdown of *ADNP* in bladder cancer cell lines significantly reduced cell proliferation and migration, which in turn, increased cisplatin resistance. Moreover, *ADNP* was found to be associated with cisplatin resistance in bladder cancer in vivo. In contrast, the upregulation of *ADNP* yield the opposite effect. These findings suggest that *ADNP* accelerates cell migration, promotes EMT, and increases cisplatin resistance in bladder cancer. In addition, *ADNP* also activates the TGF-ß/Smad signaling pathway [136].

*ADNP* is highly expressed in bladder cancer, wherein the mRNA and protein expression of *ADNP* was significantly upregulated in bladder cancer tissues as compared to normal bladder tissues. In addition, ADNP protein expression was higher in MIBC and high-grade tumors than in NMIBC and low-grade tumors. Patients with high ADNP expression had lower overall survival rates than that of patients with low ADNP expression. High ADNP protein expression was significantly associated with poor prognosis of bladder cancer and an increased risk of mortality. The knockdown of *ADNP* markedly reduced bladder cancer cell proliferation in vitro, the growth of bladder cancer in vivo, and G1/S phase transition of the cell cycle. In contrast, the overexpression of *ADNP* yielded the opposite effect. These findings suggest that *ADNP* increased bladder cancer cell proliferation by accelerating the G1/S phase transition of the cell cycle. Hence, *ADNP* might play a key role in bladder tumorigenesis. At a molecular level, *ADNP* was found to activate the AKT/*MDM2*/*p53* signaling pathway, thus promoting bladder cancer cell proliferation. Taken together, *ADNP* is overexpressed in bladder cancer and may act as an oncogene in bladder tumorigenesis [137]. Hence, *ADNP* may be a novel molecular target for predicting prognosis in bladder cancer as well as adjuvant therapeutic for bladder cancer patients receiving intravesical chemotherapy.

#### 3.10.2. *ZEB1*

*ZEB1* mRNA expression was significantly higher in bladder cancer tissues than in adjacent normal tissues. *ZEB1* overexpression was associated with greater tumor size. These findings suggest that *ZEB1* plays a role in bladder cancer and may potentially be a biomarker for the early detection and progression of bladder cancer [138]. In a study using publicly available RNA sequencing (RNA-seq) data, high ZEB1 protein expression was related to poor survival in bladder cancer patients. ZEB1 expression was also found to be clinically relevant. *ZEB1* knockdown in bladder cancer T24 cells resulted in a significant reduction in cell proliferation when compared to the control cells. These finding suggest that *ZEB1* may play an oncogenic role in bladder cancer [139].

Higher ZEB1 expression was found in MIBC and high-grade tumors as compared to NMIBC and low-grade tumors. Both the mRNA and protein expression of *ZEB1* were higher in bladder cancer UMUC3 and J82 cells as compared to the SV-HUC-1 cells. The downregulation of *ZEB1* inhibited the formation of vasculogenic mimicry, whereas the overexpression of *ZEB1* was significantly positively associated with vasculogenic mimicry. These findings suggest that *ZEB1* plays an important role in the process of vasculogenic mimicry formation in bladder cancer [140].

Six differentially expressed mRNAs (DEmRNAs) comprising of *ZEB1*, *AIFM3*, *DUSP2*, *JUN*, *MAP1B*, and *TMEM100* were found to be significantly associated with the overall survival of bladder cancer patients. The six DEmRNAs were identified as independent prognostic factors for overall survival and were associated with the pathogenesis of bladder cancer [141]. In another study, seven ferroptosis-related genes (*ZEB1, G6PD*, *PRDX6*, *SCD*, *SLC38A1*, *SRC*, and *TFRC*) were identified as a prognostic signature for bladder cancer. The prognostic signature had high accuracy in predicting the overall survival of bladder cancer patients [142]. In another study, eight ferroptosis-related genes (*ZEB1*, *ISCU*, *JDP2*, *MAFG*, *NFE2L2*, *SCD*, *TXNIP*, and *VDAC2*) were established as a prognostic model. *ISCU*, *NFE2L2*, and *TXNIP* were classified as low-risk genes, while *ZEB1*, *JDP2*, *MAFG*, *SCD*, and *VDAC2* were classified as high-risk genes. These genes may be reliable prognostic biomarkers for bladder cancer [143].

Six invasive bladder cancer cells (J82, KK47, KU7, T24, TCCSUP, and UMUC3) that exhibit mesenchymal morphology were found to have high ILK protein expression, high *ZEB1* expression, and low E-cadherin expression. These findings indicate that *ZEB1* plays a crucial role in the regulation of E-cadherin expression in bladder cancer. The overexpression of ILK-induced GSK3β resulted in E-cadherin suppression and promoted cell invasion. However, the knockdown of *ILK* suppressed cell invasion through the regulation of E-cadherin and *MMP9*. These findings suggest that the ILK–GSK3β–ZEB1 pathway is important in regulating the EMT in bladder cancer through the regulation of E-cadherin and potentially other pathways associated with ILK regulation [144].

Urothelial and adjacent sarcomatoid morphologies of MIBC arise from the same common ancestor and share a basal-like phenotype. When shifting from the urothelial to the sarcomatoid morphology, *ZEB1* and *TWIST1* expression was found to be increased while the expression of E-cadherin decreased. The divergence between the morphologies at the genome, transcriptome, and proteome levels suggest differential sensitivity to therapies [145]. In another study, a higher mRNA expression of *ZEB1*, *PD-L1*, *TIMP2*, *TWIST1*, and *VIM* was detected in NMIBC patients with pT1 tumors as compared to those with pTa tumors. A strong association was found between *PD-L1* and *TIMP2*/*TWIST1* as well as between *TIMP2* and *ZEB1*/*VIM*. High *ZEB1*, *PD-L1*, *TIMP2*, *TWIST1*, and *VIM* expression represents a specific gene signature in blood-circulating tumor cells from NMIBC patients. In addition, a significant correlation was found between high *ZEB1*, *PD-L1*, *TIMP2,* and *TWIST1* expression, and a reduced recurrence-free survival. These findings demonstrate that NMIBC patients with high *ZEB1*, *PD-L1*, *TIMP2,* and *TWIST1* expression tend to have a worse prognosis. Therefore, it is necessary to consider these patients as candidates for systemic therapy approaches with immune checkpoint inhibitors [146].

The upregulation of cytokeratin 18 and cytokeratin 19 (epithelial markers) and the downregulation of vimentin, N-cadherin, *MMP2*, and *ZEB1* (mesenchymal markers) were observed in the bone metastatic T24-B bladder cancer cells. In addition, the T24-B cells also displayed increased adhesion but decreased cell invasion or migration abilities. This suggests that the T24-B cells was able to reacquire their epithelial phenotypes after metastasizing to the bone. The PI3K/Akt pathway targets GSK3β/β–catenin to regulate *ZEB1* transcription and subsequently regulates the expression of cytokeratins, vimentin, and *MMP2* for tumor cell adhesion, invasion, and migration. The overexpression of *ZEB1* in the T24-B cells abrogates cytokeratin 18 and cytokeratin 19 expression, increases vimentin and *MMP2* expression. These findings suggest that *ZEB1* plays important roles in bladder cancer cell adhesion, migration, invasion, and distant metastasis, thus suggesting that *ZEB1* is a potential prognostic biomarker and therapeutic target for metastatic bladder cancer [147].

The expression of ZEB1 was higher in bladder cancer tissues than in normal bladder tissues. ZEB1 was negatively correlated with the expression of GRHL2. Upregulation of GRHL2 was found to directly inhibit the expression of ZEB1. The overexpression of *ZEB1* reduced the mRNA expression level of E-cadherin and promoted the expression levels of vimentin, Snail, and Slug as compared to that of the control group. Taken together, these findings suggest that *ZEB1* is the downstream target of *GRHL2*, whereby *GRHL2* inhibits the EMT process by targeting *ZEB1* [148].

The expression of ZEB1 and HIF-1α in bladder cancer tissues was significantly higher than in normal bladder tissues. In addition, the expression of ZEB1 and HIF-1α were significantly increased in high-grade, invasive, and metastatic bladder cancer tissues as compared to the low-grade, superficial, and non-metastatic bladder cancer tissues. A significant positive association was found between ZEB1 and HIF-1α protein expression in bladder cancer. The knockdown of *HIF-1α* significantly increased the expression of *ZEB1* and promoted cell migration, invasion, and EMT. These findings suggest that *HIF-1α* plays an important role in bladder cancer metastasis, and both *HIF-1α* and *ZEB1* may be potential therapeutic targets for inhibiting bladder cancer metastasis [149].

*ZEB1* is a direct target of miR-23b. A significant reduction in the protein level of *ZEB1* was observed after miR-23b overexpression, thus indicating that post-transcriptional regulation of *ZEB1* can be modulated through the targeting of its 3′ UTR. The overexpression of miR-23b resulted in the suppression of *ZEB1* in bladder cancer cells. Meanwhile, the knockdown of *ZEB1* decreased bladder cancer cell migration and invasion. These findings suggest that miR-23b directly targets *ZEB1* in bladder cancer and has diagnostic and prognostic significance [150]. Apart from miR-23b, miR-429 was also found to inhibit bladder cancer cell migration and invasion. In addition, miR-429 was able to reduce *ZEB1* expression, restore E-cadherin expression, and downregulate ß-catenin. These findings suggest that miR-429 potentially inhibits EMT by targeting *ZEB1* and ß-catenin [151].

Ginkgolide B (an anticancer drug) decreased ZEB1 protein in a dose-dependent manner and resulted in the suppression of bladder cancer cell invasion. However, the overexpression of *ZEB1* abolished the ginkgolide-B-induced suppression of bladder cancer cell invasion. The overexpression of miR-223-3p decreased ZEB1 protein production, which resulted in a decrease in bladder cancer cell invasion. In contrast, the depletion of miR-223-3p yielded the opposite effect. These findings suggest that ginkgolide B inhibits bladder cancer cell invasion through the suppression of ZEB1 protein translation via the upregulation of miR-223-3p [152]. A high DDR1 expression in bladder cancer was also found to be correlated with poor prognosis of bladder cancer. The overexpression of *DDR1* promotes the cell invasion of bladder cancer in vitro and tumor xenograft growth in vivo, whereas the knockdown of *DDR1* yielded the opposite effect. Moreover, DDR1 increased the protein levels of ZEB1 and Slug. These findings suggest that *DDR1* enhances bladder cancer cell invasion through the regulation of *ZEB1* and *Slug* expression [153].

In highly metastatic T24-L cells, silibinin inhibited GSK3ß/ß-catenin signaling and *ZEB1* expression. On the other hand, ß-catenin regulates CSC properties via *ZEB1* in bladder cancer cells. The knockdown of *ZEB1* could inhibit the mRNA, protein, and promoter activity of *CD44* in the T24-L cells. Silibinin may targets CSC phenotype by inhibiting *ZEB1* and its downstream *CD44* expression. These findings suggest that silibinin inhibited *CD44* expression and CSC properties through *ZEB1*. Consistent with the in vitro findings, silibinin was shown to suppress ß-catenin/*ZEB1* signaling, EMT, and CSC properties in vivo. Taken together, silibinin inhibits ß-catenin/*ZEB1* signaling and suppresses bladder cancer metastasis via dual-blocking EMT and stemness [154].

Sulforaphane (SFN), a chemopreventive agent that is abundant in broccoli and broccoli sprouts. SFN was found to inhibit the migration and invasion in bladder cancer T24 cells. The miR-200c inhibitor can reverse the inhibition of *ZEB1* and the induction of E-cadherin via SFN. Two transcriptional repressors, which are *ZEB1* and Snail—that negatively and transcriptionally regulate E-cadherin expression—were activated, eventually blocking the EMT process. *COX-2*, *MMP2*, *MMP9*, and E-cadherin are involved in the inhibitory effect of SFN against metastasis. In addition, the miR200c/*ZEB1* pathway is also involved in the EMT regulation by SFN. Collectively, these findings suggest that SFN suppresses bladder cancer metastasis via the *COX-2*/*MMP2, MMP9*/*ZEB1,* and Snail pathway as well as the miR-200c/*ZEB1* pathway [155].

IncRNA *ZEB1-AS1* expression was found to be significantly increased in MIBC tissues. In addition, *ZEB1-AS1* expression is associated with bladder cancer metastasis. The knockdown of *ZEB1-AS1* inhibited the migration and invasion in bladder cancer cells. The tumors formed by the *ZEB1-AS1* knockdown of bladder cancer cells grown in nude mice exhibited sharp edges. In contrast, the control tumors exhibited spike-like structures that invaded the surrounding muscle tissues. Taken together, these findings indicate that *ZEB1-AS1* regulates bladder cancer cell migration, invasion in vitro, and its metastasis in vivo, thus suggesting that *ZEB1-AS1* may be an oncogene in bladder cancer. *ZEB1-AS1* regulates bladder cancer metastasis through the upregulation of ZEB1 protein. On the other hand, *ZEB1-AS1* increased ZEB1 protein expression by recruiting AUF1 to activate the translation of *ZEB1* mRNA. The elucidation of the role of *ZEB1-AS1* in the progression of bladder cancer will improve the understanding of lncRNA-induced tumorigenesis and metastasis in bladder cancer [156].

#### 3.10.3. *ZEB2*

ZEB2 expression was significantly higher but miR-138 was significantly lower in bladder cancer tissues when compared to normal bladder tissues. There was a strong inverse correlation found between ZEB2 and miR-138. *ZEB2* is a target of miR-138, wherein miR-138 targets the 3′ UTR of the *ZEB2* mRNA, thus leading to the inhibition of its protein translation. The overexpression of miR-138 significantly increased E-cadherin and decreased ZEB2 protein and vimentin. In contrast, the depletion of miR-138 significantly decreased E-cadherin and increased ZEB2 protein and vimentin. Moreover, the overexpression of miR-138 decreased migration and invasion, while the depletion of miR-138 yielded the opposite effect. The suppression of *ZEB2* abolished the effects of miR-138 on E-cadherin, vimentin, cell migration, and invasion. These findings suggest that miR-138 inhibits bladder cancer cell migration and invasion through *ZEB2* suppression. The miR-138/*ZEB2* regulatory axis may play an important role in the regulation of bladder cancer cell migration and invasion [157]. In addition, ZEB2 was found to be upregulated in bladder cancer tissues and higher ZEB2 expression was associated with a lower overall survival rate. *ZEB2* is also a target of miR-454-3p and miR-374b-5p. Both miR-454-3p and miR-374b-5p are able to inhibit cell migration, invasion, and EMT in bladder cancer by targeting *ZEB2* [158].

*ZEB2* is also a downstream target of miR-145 that represses ZEB2 protein expression in bladder cancer cells. The expression of *TUG1* was found to be upregulated in bladder cancer tissues and cell lines. *TUG1* elevated *ZEB2* expression by negatively regulating miR-145 expression. *TUG1* promoted bladder cancer cell invasion and radiotherapy resistance by inducing EMT [159]. In another study, both the mRNA and protein expression of *TUG1* and *ZEB2* were significantly increased in bladder cancer tissues as compared to normal bladder tissues. These findings suggest that *TUG1* and *ZEB2* might play an oncogenic role in bladder cancer. The knockdown of either *ZEB2* or *TUG1* inhibited cell proliferation and induced apoptosis in bladder cancer. The overexpression of *ZEB2* reversed the effects of *TUG1* knockdown on cell proliferation and apoptosis. *TUG1* binds to miR-142 to regulate *ZEB2* expression. The knockdown of *TUG1* suppressed the activation of the Wnt/ß-catenin pathway by affecting *ZEB2* expression [160].

MiR-377 targets *ZEB2* and suppress its expression in bladder cancer cells. CircZFR directly binds to miR-377 as a sponge to promote *ZEB2* expression. The knockdown of circZFR inhibits cell proliferation and migration by targeting the miR-377/*ZEB2* axis. Taken together, these findings suggest that circZFR promotes bladder cancer progression by regulating miR-377/*ZEB2* signaling and circZFR could be a therapeutic marker in bladder cancer [161].

The expression of *ZEB2-AS1* was found to be significantly increased in bladder cancer tissues as compared to adjacent normal tissues. The high *ZEB2-AS1* expression significantly correlated with tumor size, tumor stage, and lymph node metastasis. The *ZEB2-AS1* expression levels in bladder cancer cell lines were significantly higher as compared to that of bladder cancer cells. These findings indicate that *ZEB2-AS1* was upregulated in bladder cancer tissues and cell lines. *ZEB2-AS1* knockdown repressed the proliferation and induced apoptosis of bladder cancer cells. In contrast, *ZEB2-AS1* overexpression markedly increased cell viability and inhibited apoptosis. MiR-27b was downregulated in bladder cancer tissues as compared to adjacent normal tissues. *ZEB2-AS1* was significantly negatively correlated with miR-27b expression in bladder cancer tissues, thus indicating an inverse correlation relationship between *ZEB2-AS1* and miR-27b. The knockdown of *ZEB2-AS1* promoted miR-27b expression in bladder cancer cells, whereas *ZEB2-AS1* overexpression dramatically reduced miR-27b expression. MiR-27b knockdown significantly reversed si-*ZEB2-AS1*-mediated inhibition on bladder cancer cell proliferation and almost eliminated the pro-apoptotic effect of si-*ZEB2-AS1* in bladder cancer cells. Taken together, these findings suggest that *ZEB2-AS1* promotes bladder cancer tumorigenesis through the downregulation of the tumor-suppressive miR-27b [162].

The protein expression of ZEB1 and ZEB2 were detected in 7.5% and 24% of bladder cancer tissues, respectively. In contrast, neither ZEB1 nor ZEB2 protein expression were detected in the normal bladder tissues. Nuclear ZEB1 expression was detected in 22.8% of NMIBCs and in 21.7% of MIBCs. In addition, ZEB1 protein expression was also found in invasive bladder cancer cell lines (EJ, HU456, J82, KK47, MGHU1, T24, and UMUC3) [163]. The majority of NMIBCs showed no or low nuclear ZEB1 staining, whereas strong nuclear ZEB1 staining was observed in the majority of MIBCs. There were relationships between ZEB1 and ZEB2 expression, prognosis, and clinical outcomes of bladder cancer. ZEB1 staining was significantly increased in higher stage tumors [147]. Bladder cancer patients with ZEB2-immunopositive tumors had a lower 5-year survival and most likely died from the disease as compared to those with ZEB2-immunonegative tumors. The findings suggest that ZEB2 could be a predictor of cancer-specific survival [164].

The forced expression of *ZEB1* enhances invasion potential, whereas the knockdown of *ZEB1* reduces cell migration and invasion potential, thus suggesting that *ZEB1* promotes cell migration and invasion in bladder cancer [163]. On the other hand, *ZEB2* strongly decreases UV-induced DNA fragmentation, thus suggesting that *ZEB2* protects bladder cancer cells from DNA damage-induced apoptosis. Hence, *ZEB2* may contribute to tumor progression by protecting bladder cancer cells from apoptosis. A high mRNA and protein expression levels of *ZEB1* and *ZEB2* were observed in mesenchymal bladder cancer cells (J82, T24, and UMUC3), whereas low or absent *ZEB1* and *ZEB2* expression was observed in bladder cancer cells (HT1376, RT4, and RT112). The findings indicate that *ZEB1* and *ZEB2* are highly expressed in bladder cancer cell lines that exhibit mesenchymal phenotypes, thus indicating their involvement in the EMT of bladder cancer [164]. In another study, NMIBC was found to have higher expression levels of epithelial markers (E-cadherin and p63), while MIBC was found to have higher expression levels of mesenchymal markers (*ZEB1*, *ZEB2*, *MMP2*, *MMP9*, and vimentin). Interestingly, a subset of MIBCs maintained high expression levels of E-cadherin and p63, thus suggesting that the epithelial phenotype remains present in a subset of MIBCs [165].

The miR-200 family (miR-141, miR-200a, miR-200b, miR-200c, and miR-429) expression was inversely correlated with the expression ZEB1 and ZEB2. The ZEB1 and ZEB2 expression was inversely correlated with the E-cadherin expression. These findings suggest that the miR-200 family members inhibits EMT in bladder cancer cells by targeting the transcriptional repressors of E-cadherin (*ZEB1* and *ZEB2*) [166]. The miR-200 family expression was increased in NMIBC as compared to normal bladder tissues but its expression was found to be reduced in MIBCs as compared to NMIBCs. In MIBCs, both the *ZEB1* and *ZEB2* expression were reduced and negatively correlated with the miR-200 family expression. These findings indicate that the miR-200 family potentially regulates the EMT in MIBC progression by regulating *ZEB1* and *ZEB2* expression [167]. The relationship between miR-200 family and *ZEB1*/*ZEB2* was also observed when lncRNA *SNHG16* was found to promote EMT by increasing *ZEB1* and *ZEB2* expression via the targeting of miR-200a-3p. These findings suggest that *SNHG16* promotes EMT through the miR-200a-3p/*ZEB1*/*ZEB2* axis [168].

The miR-205 was able to distinguish NILGC from NIHGC, with a sensitivity of 95.8% and a specificity of 96.7%. The miR-145 distinguished NIHGC from infiltrating carcinoma, with a sensitivity of 100% and a specificity of 91.7%. The miR-125b expression was significantly lower in NILGC than in PUNLMP, with a sensitivity of 93.3% and a specificity of 84.2%. The ZEB1 and ZEB2 expression were associated with tumor grade and miRNA expression. ZEB1 immunoreactivity was more frequently detected in NIHGC than in NILGC as well as in infiltrating carcinoma. In contrast, ZEB2 immunoreactivity was more frequent in infiltrating carcinoma than in NIHGC. These findings suggest that ZEB1/ZEB2 expression can be used to distinguish between different grades of papillary urothelial carcinomas. Hence, ZEB1 and ZEB2 may be useful as a complementary diagnostic biomarkers for the grading or classification of bladder cancer [169].

IncRNA *ZFAS1* is overexpressed in bladder cancer tissues and cell lines. The knockdown of *ZFAS1* showed reduced cell migration and invasion in bladder cancer. In addition, reduced *ZFAS1* markedly decreased *ZEB1*, *ZEB2*, and vimentin expression levels but increased *KLF2*, *NKD2*, and E-Cadherin expression levels. These findings demonstrate that *ZFAS1* knockdown potentially inhibits cell migration and invasion by downregulating *ZEB1* and *ZEB2* expression and inhibits cell proliferation by upregulating *KLF2* and *NKD2* expression [170].

#### 3.10.4. *TSHZ3*

The expression levels of *TSHZ3*, *ISL*, *MEIS1*, *ZEB2*, and *ZFHX4* were significantly lower, whereas the expression of *HOXC4* was higher in bladder cancer tissues when compared to that of normal bladder tissues. Five out of the six homeobox genes (*TSHZ3*, *ISL1*, *MEIS1*, *ZEB2*, and *ZFHX4)* had a strong correlation with most cytokines, while *HOXC4* was found to have a strong correlation with *IL-17A*. In addition, the expression of the six signature genes correlated with most immune checkpoints (CTLA-4, PD-1, PD-L1, and PD-L2). The developed prognostic signature showed good accuracy and consistency in predicting prognosis and response to immunotherapy. Hence, the prognostic signature could be a potential biomarker and therapeutic target for bladder cancer [171].

#### 3.10.5. *ZHX3*

*ZHX3* was found to be highly expressed in bladder cancer tissues and cell lines, thus indicating that *ZHX3* is upregulated in bladder cancer. High ZHX3 expression was positively correlated with worse clinical outcomes, such as advanced T stage, N stage, and recurrence. Bladder cancer patients with high ZHX3 expression had poorer disease-free survival and shorter overall survival rates. Thus, *ZHX3* is an independent prognostic factor of bladder cancer. The overexpression of *ZHX3* in bladder cancer T24 cells significantly increased cell migration and invasion. However, the knockdown of *ZHX3* in SV-HUC-1 cells did not affect cell migration and invasion. *ZHX3* is a downstream target gene of *TRIM21* that regulates the stability of ZHX3 protein by modulating through proteasomal degradation. RNA-seq analysis revealed that when *ZHX3* was downregulated, *RGS2* was upregulated, whereas, when *ZHX3* was upregulated, *RGS2* was downregulated. These findings indicate that *RGS2* is the downstream target gene of *ZHX3* in bladder cancer. The knockdown of *RGS2* markedly restored the abilities of migration and invasion in the *ZHX3* knockdown bladder cancer cells. These findings support the important regulatory role of the *ZHX3*/*RGS2* axis in mediating metastasis. In bladder cancer tissues, *ZHX3* expression negatively correlated with *RGS2* expression. Bladder cancer patients with high ZHX3 and low RGS2 expression had the worst prognosis. The knockdown of *ZHX3* reduced the expression of *RhoA*. Moreover, *RhoA* activity was decreased in *ZHX3*-knockdown bladder cancer cells but was significantly increased in *ZHX3*-knockdown bladder cancer cells treated with siRGS2. These findings suggest that *ZHX3* acts as an oncogene to promote bladder cancer cell aggressiveness through the *RGS2*/*RhoA* pathway. Thus, *ZHX3* could be used as a prognostic biomarker for bladder cancer [172].

## 4. Conclusions

Although there are several commercially available biomarkers for bladder cancer, none of them can completely replace cystoscopy, which is the gold standard for the diagnosis and surveillance of bladder cancer. Therefore, the need for developing new non-invasive biomarkers for bladder cancer remains crucial. New approaches focusing on transcriptomic dysregulation of special interest since the changes in gene expression happen at a molecular rather than cellular levels and have biological and functional significance. Dysregulated homeobox genes are promising biomarkers for the diagnosis, prognosis, and therapy of bladder cancer, as they are directly associated with bladder tumorigenesis and clinically relevant bladder cancer outcomes. This review provides a snapshot summary of the homeobox genes that have been reported to be dysregulated and associated with bladder cancer. This preliminarily screening of the existing literature presents a rich list of homeobox genes for further biomarker evaluation by urologic researchers and clinicians. We tried to simplify the current landscape of homeobox gene dysregulation in bladder cancer in order to enrich the knowledge of researchers and clinicians involved in the development of new non-invasive biomarkers. It is hoped that this will contribute towards the effective clinical translation of homeobox-gene-derived biomarkers; thereby relieving the human and economic costs associated with unoptimized treatment of patients and ultimately, improving the survival of bladder cancer patients through earlier and more accurate detection and novel therapeutic modalities.

## Figures and Tables

**Figure 1 diagnostics-13-02641-f001:**
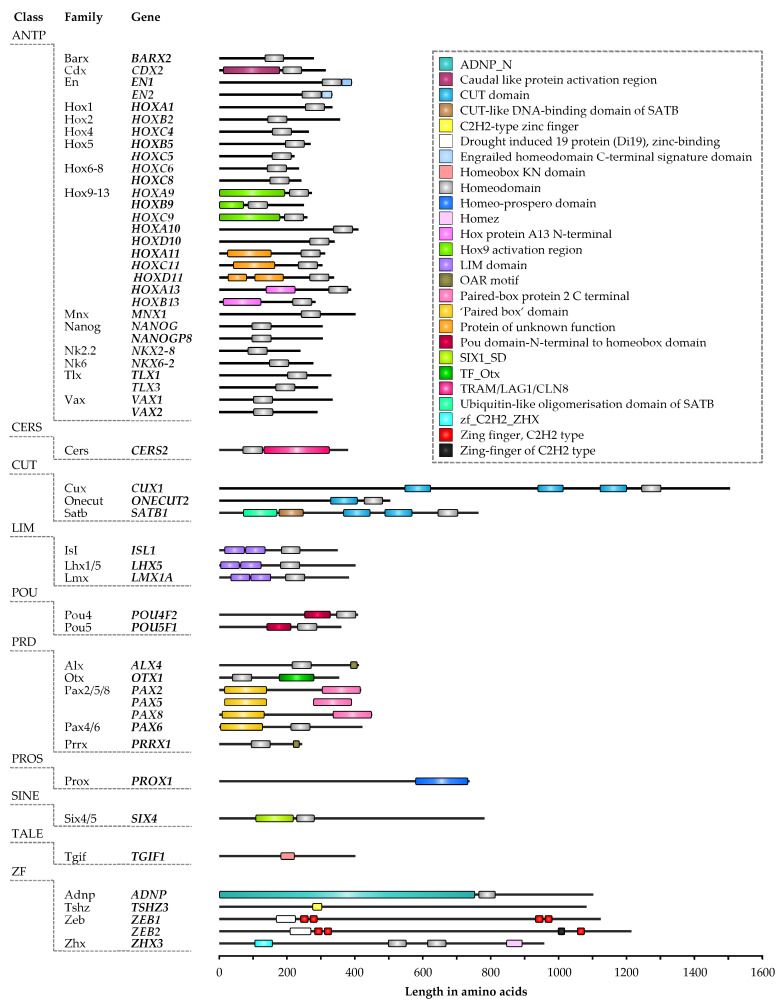
Homeodomain and specific protein domains of the human homeobox genes reported to be dysregulated in bladder cancer.

**Figure 2 diagnostics-13-02641-f002:**
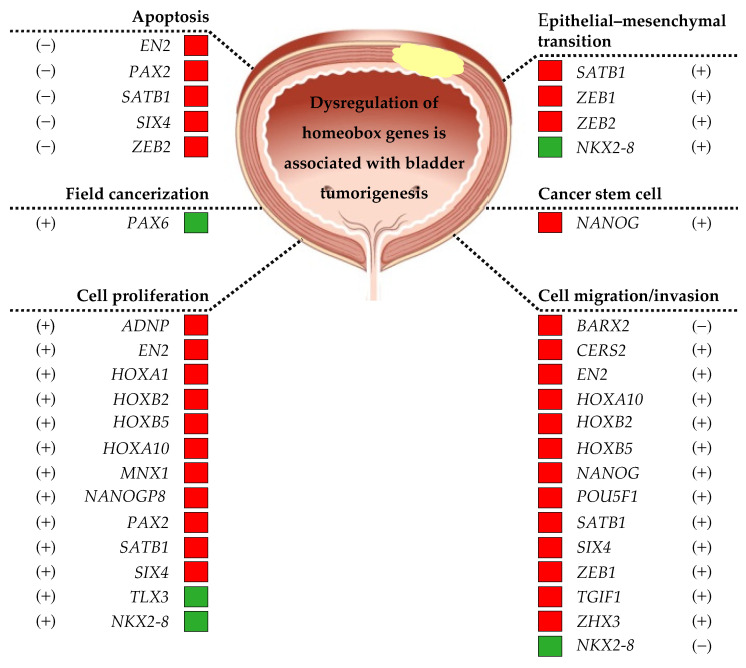
Dysregulation of homeobox gene expression in bladder cancer is associated with bladder tumorigenesis. Homeobox genes that are upregulated and downregulated in bladder cancer are highlighted in red and green boxes, respectively. The dysregulated expression of homeobox genes led to an increase (+) or a decrease (−) in specific biological activities in bladder cancer.

**Figure 3 diagnostics-13-02641-f003:**
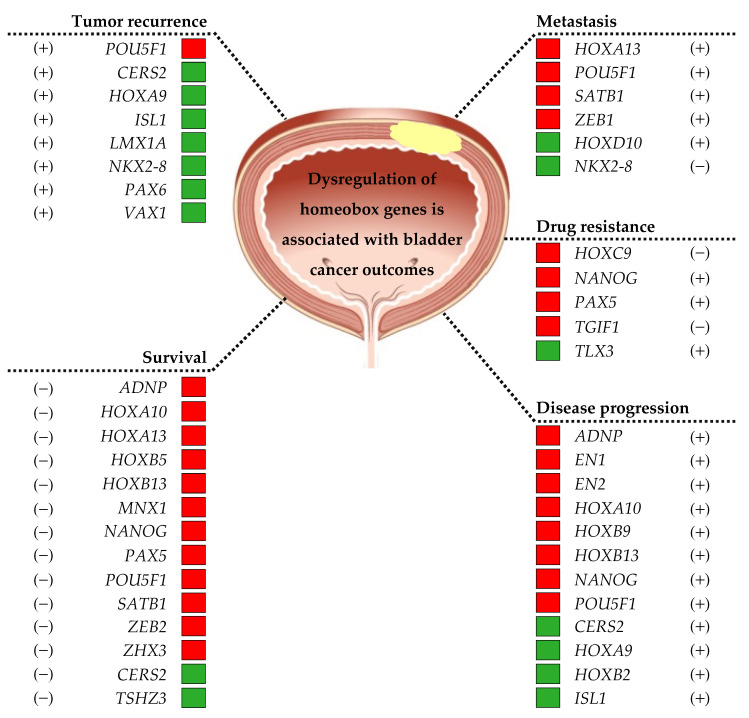
Dysregulation of homeobox gene expression in bladder cancer is associated with clinically relevant bladder cancer outcomes. Homeobox genes that are upregulated and downregulated in bladder cancer are highlighted in red and green boxes, respectively. Plus sign (+) indicates an increased risk of tumor recurrence, metastasis, disease progression, drug resistance, and patient survival. Meanwhile, minus sign (−) indicates a decreased risk of tumor recurrence, metastasis, disease progression, drug resistance, and patient survival.

## Data Availability

Data sharing is not applicable to this article.

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
