# Peer review of "Homeobox Gene Expression Dysregulation as Potential Diagnostic and Prognostic Biomarkers in Bladder Cancer"

_diagnostics, 2023, doi:10.3390/diagnostics13162641_

Round 1
Reviewer 1 Report
Dear Authors,
Review article on Homeobox genes as biomarkers of bladder cancer provides a comprehensive and well-structured analysis of homeobox genes and their dysregulation in bladder cancer. This review collects all the information of the research conducted on homeobox genes and thus significantly contributes to our understanding of the molecular mechanisms underlying bladder cancer progression. The introduction effectively sets the stage by providing a concise background on bladder cancer and highlighting the importance of biomarkers in its diagnosis and management. This review demonstrates a strong command of the subject matter and exhibits a high level of scientific rigor.
Author Response
Thank you for reviewing our manuscript.
Reviewer 2 Report
The article is devoted to an urgent topic related to the search for potential markers of the development of bladder cancer. Extensive analytical studies have been carried out, a high level of systematization of the data obtained is noted. However, the conclusion is poorly written, there is no practical significance of the review. What does this review contribute to the planning of experimental, preclinical and clinical studies?
The article is devoted to an urgent topic related to the search for potential markers of the development of bladder cancer. Extensive analytical studies have been carried out, a high level of systematization of the data obtained is noted. However, the conclusion is poorly written, there is no practical significance of the review. What does this review contribute to the planning of experimental, preclinical and clinical studies?
